# Synthesis of Ni-Cu Heterostructures with SPS to Achieve a Balance of Strength and Plasticity

**Wuqiang Ai [1], Zewen Yu [1] and Yaojun Lin [1,2,*]**

[1] School of Materials Science and Engineering, Wuhan University of Technology, Wuhan 430070, China; ai688awq@163.com (W.A.); ze2073261@163.com (Z.Y.)

[2] State Key Laboratory of Advanced Technology for Materials Processing and Synthesis, Wuhan University of Technology, Wuhan 430062, China

* Correspondence: yjlin@whut.edu.cn

**Abstract:** The balance between strength and plasticity has always been an urgent problem for researchers to solve. In this experiment, Ni-Cu heterostructures (HSs) were synthesized by spark plasma sintering (SPS), rolling deformation, and subsequent heat treatment. The density of the Ni/Cu interface of Ni-Cu HS materials can be independently tuned, and thus the effect of hetero-deformation-induced (HDI) strengthening in Ni-Cu heterostructures can be tuned. The density of the Ni/Cu interface is tuned by adding Cu with different volume fractions to obtain the best combination of strength and plasticity. Compared with the previous HSs, the hardness differences between different regions of Ni-Cu HSs are more significant, and they are all composed of single substances. The hard Ni domain and the soft Cu domain are not only different in phase composition but also different in grain size. More interestingly, the density of the hard/soft domains can be adjusted independently, which provides a new way to explore the strength and plasticity balance of HS materials. The yield strength of Ni-Cu HS materials first increases and then decreases gradually with the increase in the Cu volume fraction. When the Cu volume fraction is less than 30%, the HDI strengthening effect in the Ni-Cu HS material can offset the effect of the yield strength reduction caused by Cu; with a further increase in the Cu volume fraction, the HDI strengthening effect is less than the yield strength reduction effect brought on by Cu.

**Keywords:** Ni-Cu; heterostructure materials; hetero-deformation-induced strengthening; strength; ductility

## 1. Introduction

In recent years, heterostructure (HS) materials [1] have attracted the interest of researchers due to their good balance of strength and plasticity. HS materials can be divided into the following types: gradient structure [2–6], heterogeneous lamella structure [7–9], harmonic structure [10,11], bimodal structure [12,13], dual-phase structure [14,15], etc. HSs have significant differences in the different regions, leading to unique mechanical properties. Fang et al. [5] synthesized gradient nano-grained copper coatings using a surface mechanical grinding treatment. The surface grains of the nano-copper coating gradually increased from 20 nm to 300 nm, and the yield strength of the gradient structure with a depth of 150 nm was 600 MPa, which was much higher than that of coarse-grained Cu. Wu et al. [16] used electrodeposition to uniformly distribute nano-domains in coarse-grained Ni, thus forming a nano-domain structure in which coarse-grained Ni and nano-domain Ni were alternately distributed. The nanodomain structure achieved a good balance of strength and plasticity. Zhao et al. [17] synthesized bulk multimodal and bimodal nickel by cryogenic milling and hot isostatic pressing, with a uniform elongation of 42% and 49%, and a yield strength of 457 MPa and 312 MPa, respectively. This combination of strength and ductility is far superior to nano-nickel blocks prepared by electrodeposition (EC), cold rolling, iso-channel angular extrusion, and high-pressure torsion.

Ni-Cu composite material is an important alloy commonly used in industry and marine fields [18–20]. While satisfying its strength, it is also necessary to consider that the material should have a certain plasticity. Many scholars at home and abroad have carried out extensive and in-depth research on Ni-based alloys. Guo et al. [21] used the electrodeposition method to prepare bioactive dicalcium phosphate (DCPD) coating on the surface of SLM-NiTi alloy, and the DCPD-coated SLM-NiTi alloy showed good corrosion resistance and biocompatibility. Konovalov [22] successfully deposited Ni-based superalloy powder on a titanium-based alloy substrate by selective laser melting. Compared with the substrate, the Ni-based coating exhibited excellent wear resistance, and the specific wear rate of the coating was 2.7 times lower than that of the substrate. Xiang et al. [23] successfully fabricated two medium-entropy alloy coatings (MEACs), namely, CoCrNiTi and CrFeNiTi, on pure titanium sheets by pulsed laser cladding, and MEAC showed a good combination of strength and plasticity.

The Ni regions and Cu regions are alternately and uniformly distributed in the Ni-Cu HS. The Ni regions provide the strength of the material and the Cu regions provide the necessary plasticity [8]. The superior properties of HS materials arise from hetero-deformation-induced (HDI) stress and extra strain hardening [24–26], both of which are associated with heterostructures. The heterostructure is mainly determined by (1) the strength difference between the soft and hard domains, and (2) the density of the interface of the soft and hard domains [27]. There is a significant strength difference between the hard Ni and soft Cu domains in the Ni-Cu HS. The Ni-Cu composites were subjected to rolling and subsequent heat treatment to obtain hard Ni domains and soft Cu domains with more significant differences. According to the Hall–Petch formula [28–30], the smaller the grain size, the higher the yield strength of the material. Subsequently, an appropriate heat treatment process was explored to heat the rolled samples so that the grains of the soft Cu domains were recrystallized while the grains of the hard Ni domains were basically unchanged. The difference in grain size of the hard Ni domain and the soft Cu domain resulted in a more pronounced difference in the strength of the hard Ni domain and the soft Cu domain.

## 2. Materials and Methods

Purchased commercial Ni powders with a purity >99.9% and Cu powders with a purity >99.9% were used to produce Ni-Cu HS material. First, Cu powders were mixed in a volume percent of 10, 20, 30, and 40 with Ni powders under an argon atmosphere for 1 h by a high-energy shaker mill (SPEX, 8000M Mixer/Mill, Metuchen, NJ, American) at a speed of 1080 cycles per minute. Then, the uniformly mixed powder was placed in a graphite mold with a diameter of 20 mm and pre-pressed at room temperature with a pressure of 120 MPa for 5 min. The compacted powders were consolidated via SPS in an apparatus manufactured (Shanghai Chenhua Technology Co., Ltd., Shanghai, China). During SPS, the chamber of the apparatus was first evacuated to a pressure to the order of $10^{-3}$ Pa followed by argon filling to a pressure of 7 kPa. Then, under a pressure of 50 MPa, the compacted powders were heated to 800 °C within 7 min, followed by further heating to the target temperature of 900 °C at 50 °C/min and holding at 900 °C for 5 min. Finally, the consolidated Ni-Cu materials were cooled down to room temperature in the chamber to obtain the cylindrical bulk samples. The cylindrical sample was cut into Φ20 × 5 mm and place on the rolling mill for rolling. The rolling deformation was 80% and the final thickness was 1 mm. The rolled samples were placed in a resistance furnace (KLX-12A) for annealing. The annealing temperature was 450 °C and the holding time was 1 h. The materials formed by the consolidation of the Cu powders with volume ratios of 10%, 20%, 30%, and 40% were named Ni-10Cu, Ni-20Cu, Ni-30Cu, and Ni-40Cu, respectively. The synthesis process of the Ni-Cu heterostructure materials and the instruments used are shown in Figure 1.

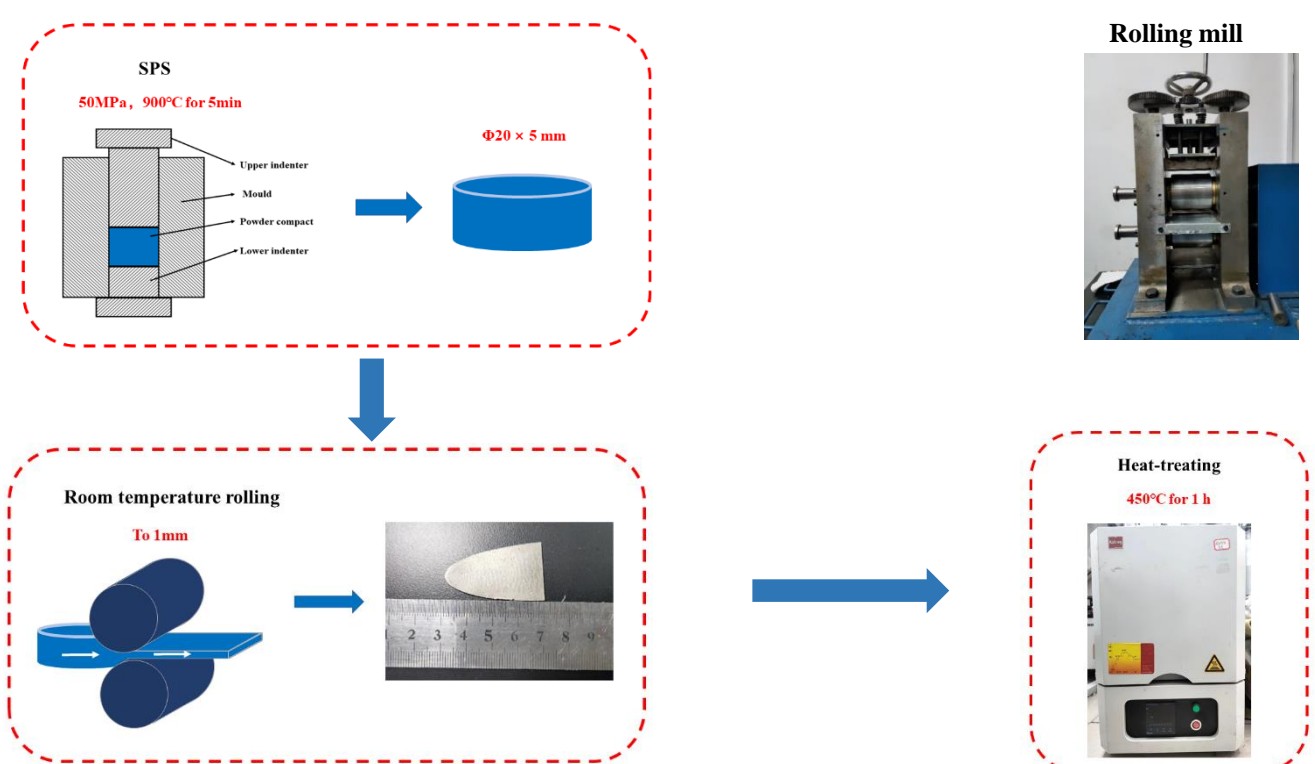

**Figure 1.** Schematic illustration of the preparation of Ni-Cu heterostructure materials by SPS and room-temperature rolling.

The consolidated Ni-Cu materials were studied using X-ray diffraction (XRD) with Cu Kα radiation operated at a voltage of 40 kV and a current of 40 mA in a Bruker D8 Advance diffractometer. Both Ni and Cu powders and the Ni-Cu HS materials were characterized by a ZEISS Gemini 300 scanning electron microscope (SEM) operated at 10 kV. The consolidated Ni-Cu HS materials were also analyzed by electron backscatter diffraction (EBSD) using an EDAX TSL Hikari EBSD detector in a FEI FEG Quanta 450 SEM with a scanning step of 0.05 μm and an acceleration voltage of 20 kV.

Both regular and LUR tensile tests were performed at room temperature with a strain rate of $1.5 \times 10^{-4}$ s$^{-1}$. The testing instrument was an Instron-5966 equipped with a video extensometer, and the unloading rate of the LUR tensile test was 200 N/min. Dog bone-shaped specimens with a standard length of 7 mm and a cross-section of 1 mm × 1 mm were used for regular and LUR tensile tests, which were cut in the rolling direction from an annealed Ni-Cu sheet and mechanically polished before testing.

## 3. Results

### 3.1. Compositions and Microstructures

The SEM secondary electron (SE) micrographs in Figure 2a,b show irregular polyhedral Ni powders with a size of 10–40 μm and spherical Cu powders with a size of 10–40 μm, respectively. Figure 3 shows the XRD curves of Ni-Cu heterostructures with different volume fractions, and the Ni-Cu heterostructures only consisted of Ni phase and Cu phase. Figure 4a–c are the SEM images and EDS energy spectra of the Ni-30Cu composites after SPS. Cu appeared to be approximately circular and uniformly distributed in the Ni matrix.

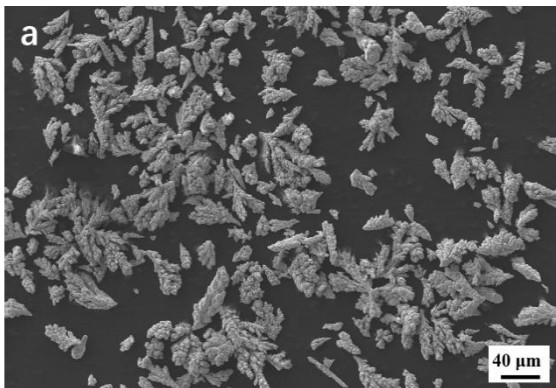
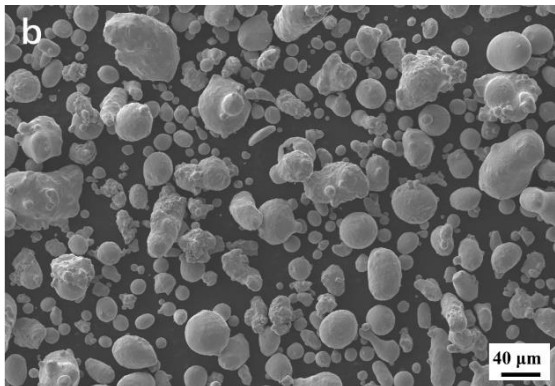

**Figure 2.** SEM images of Ni powder and Cu powder: (**a**) Ni powder; (**b**) Cu powder.

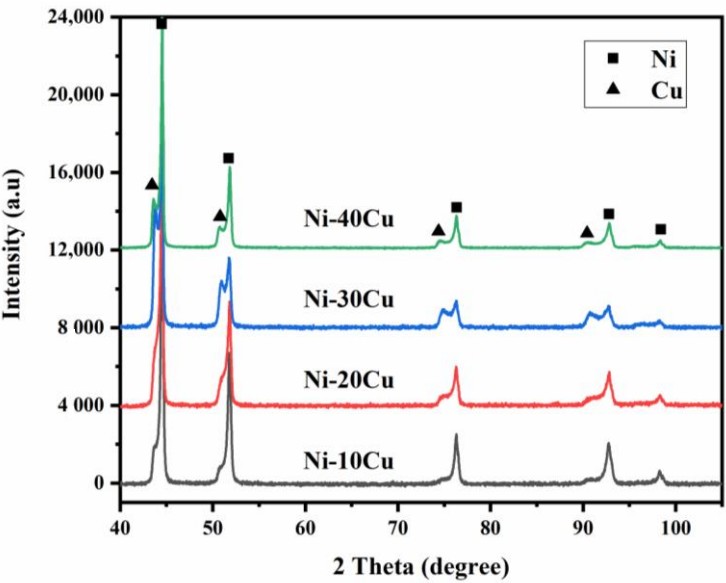

**Figure 3.** XRD curves of Ni-Cu heterogeneous materials with different volume fractions.

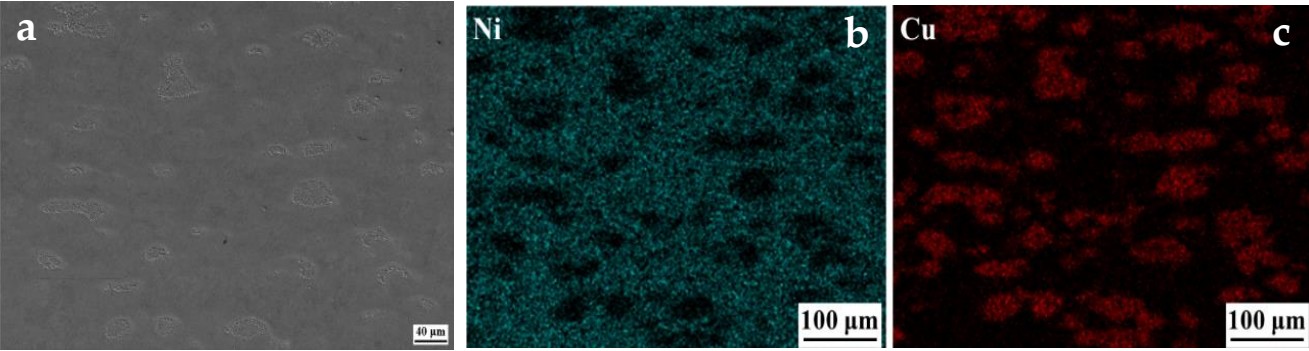

**Figure 4.** SEM morphology and EDS spectrum of Ni-30Cu as sintered: (**a**) SEM image of sintered Ni-30Cu; (**b**) distribution of Ni elements; (**c**) distribution of Cu elements.

Figure 5a–f are the SEM images and EDS spectra of Ni-Cu HS materials with different Cu contents. After rolling and subsequent heat treatment, Cu was embedded in the Ni matrix in the form of long fibers. With the increase in Cu content, the long fiber regions gradually increased, and the density of Cu regions also increased significantly. Ni domains and Cu domains were alternately arranged to form a layered structure.

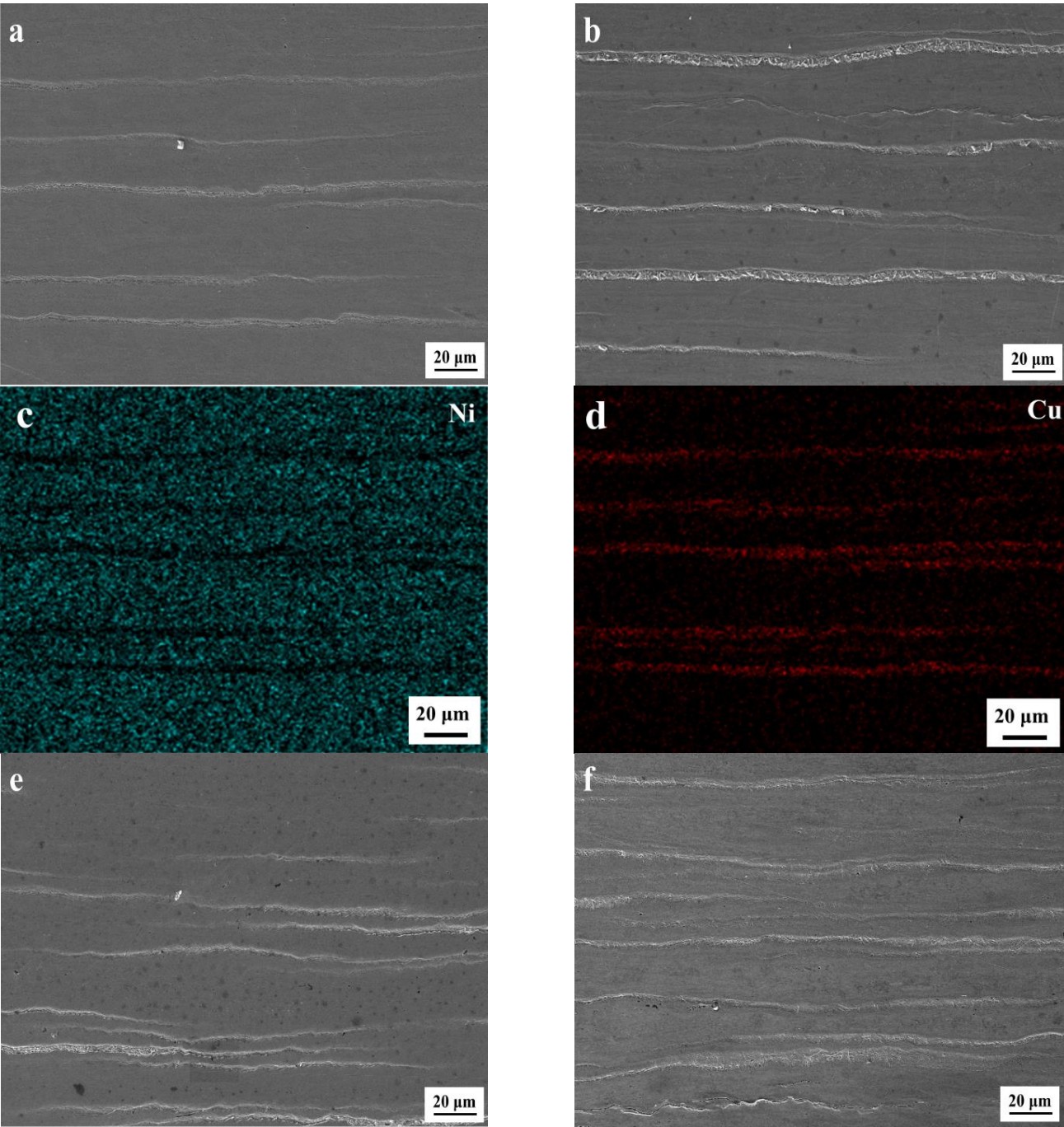

**Figure 5.** SEM images and EDS spectra of Ni-Cu heterogeneous materials with different Cu contents: (**a**) Ni-10Cu; (**b**) Ni-20Cu; (**c**) distribution of Ni element in Ni-10Cu; (**d**) distribution of Cu element in Ni-10Cu; (**e**) Ni-30Cu; (**f**) Ni-40Cu.

Figure 6a,b are the SEM images and specific numerical curves at the Ni-Cu interface, respectively. Regarding the diffusion from the Ni region to the Cu region, the longer the diffusion distance, the lesser the proportion of Ni. There was a mutual diffusion of elements at the Ni/Cu interface, and the solid solution of Ni in Cu may have increased its recrystallization temperature.

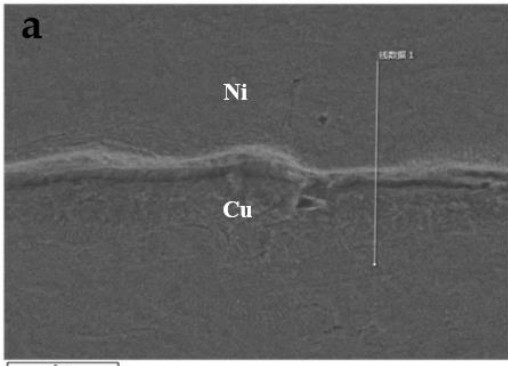
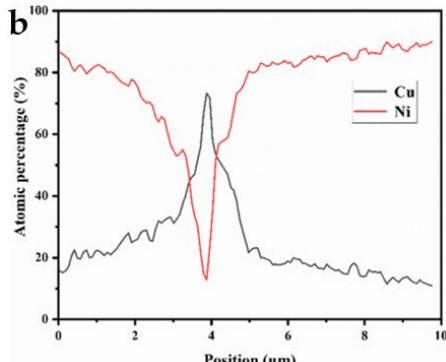

**Figure 6.** (**a**) Ni-30Cu line scan area; (**b**) line scan curve at Ni-30Cu interface.

Figure 7 shows the EBSD inverse pole figure (IPF) and phase map of Ni-30Cu as sintered, rolled, and annealed. Along the rolling direction was defined as the rolling direction (RD), and perpendicular to the rolling direction as the normal direction (ND). Figure 7a,b shows that the Ni grains and Cu grains of the sintered Ni-Cu composite were equiaxed grains, and the Ni regions and the Cu regions were evenly distributed alternately. Figure 7c,d shows that both Ni grains and Cu grains were elongated into long fibrous shapes after rolling deformation. After annealing at 450 °C for 1 h, the Cu grains gradually recrystallized into equiaxed grains, and the Ni grains remained as elongated grains. Figure 8a,b shows the average grain size statistics along the RD and the ND of Ni-30Cu as sintered state, 80% deformed rolled state, and annealed state. After rolling, the Ni grains and Cu grains were elongated along the RD, and the grains were refined along the ND. The Ni grains and Cu grains were further refined after annealing, and the Cu grains recrystallized into equiaxed grains.

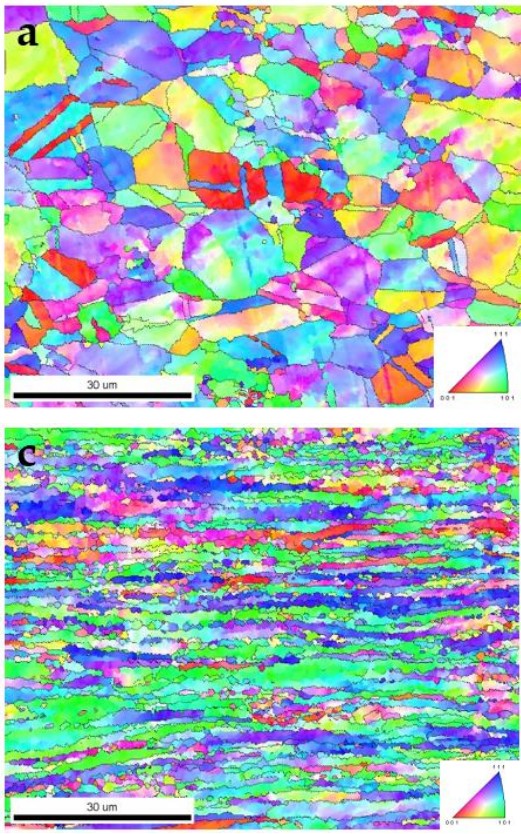
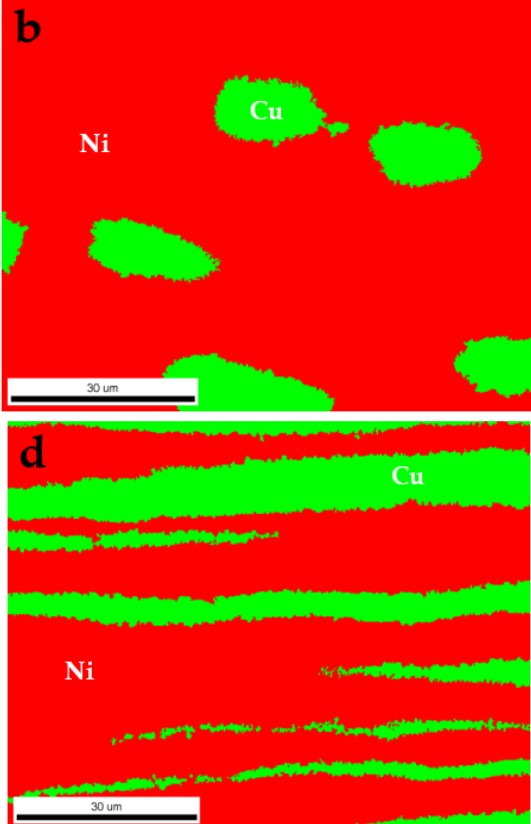

**Figure 7.** *Cont.*

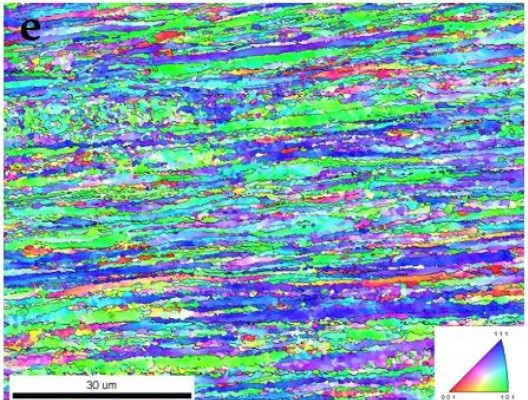
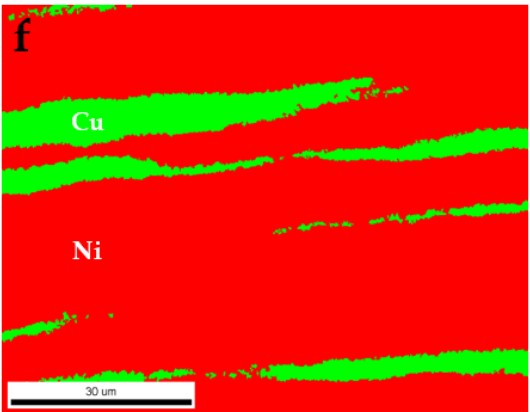

**Figure 7.** IPF and EBSD phase maps: (**a**) IPF of Ni-30Cu as sintered; (**b**) EBSD phase map of Ni-30Cu as sintered; (**c**) IPF of Ni-30Cu as rolled; (**d**) EBSD phase map of Ni-30Cu as rolled; (**e**) IPF of Ni-30Cu as annealed; (**f**) EBSD phase map of Ni-30Cu as annealed.

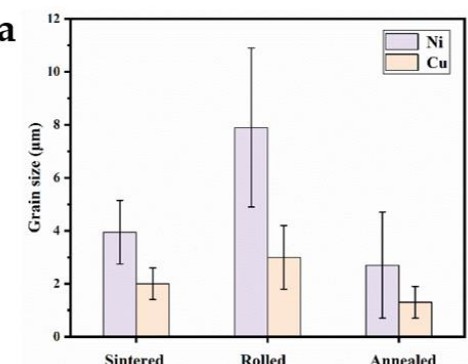
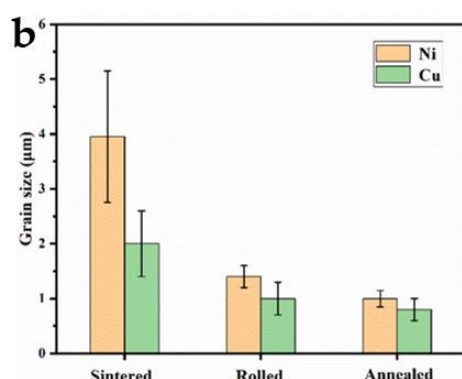

**Figure 8.** Statistics of average grain size of Ni grains and Cu grains (**a**) along the RD; (**b**) along the ND.

### 3.2. Mechanical Properties

Figure 9 shows the stress–strain curves of Ni-Cu HS materials with different Cu volume fractions. With the increase in Cu content, the yield strength of the material first increased and then decreased, and the uniform elongation also increased gradually. When the Cu volume fraction was 30%, the yield strength of the Ni-Cu HS material reached the maximum value. Ni-30Cu achieved a balance of strength and plasticity. This is mainly because the interfacial density of soft Cu domains and hard Ni domains gradually increased with the increase in Cu content, and a plastic strain gradient was generated at the Ni/Cu interface, resulting in HDI stress. The forward stress generated by HDI acted on the soft Cu domain, increasing the plastic strain of the Cu domain and increasing its ability to store dislocations. As the Cu content increased, the overall yield strength of the material decreased partially due to the addition of the soft Cu phase. The Ni/Cu domain interface did not increase linearly with Cu content. As the Cu content increased, the overall yield strength of the material decreased partially due to the addition of the soft Cu phase. The HDI enhancement in the Ni-Cu HS material could offset the effect of the decrease in yield strength, so the yield strength of Ni-30Cu was higher than that of Ni-40Cu.

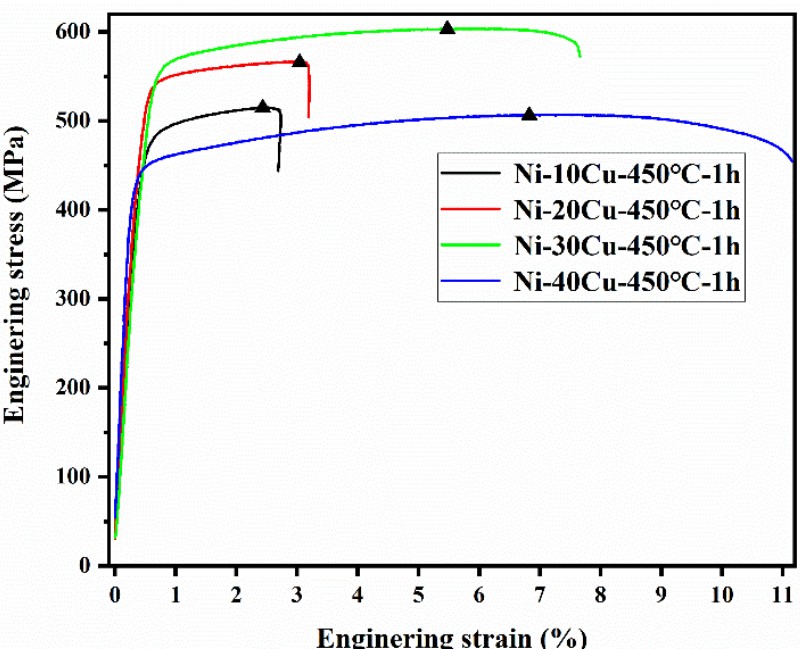

**Figure 9.** Stress–strain curves of Ni-Cu heterogeneous materials with different Cu volume fractions.

Figure 10a shows the loading, unloading, and reloading (LUR) curves of four Ni-Cu heterostructures with different Cu contents. The Ni-Cu HS materials in Figure 10b all show obvious hysteresis loops, Figure 10c is the schematic diagram of HDI stress calculation, and Figure 10d is the HDI stress calculated from the LUR stress–strain curve in (a). The HDI stress is calculated by Equations (1) and (2) [9,31,32]:

$$\sigma_{HDI} = \sigma_f - \sigma_{eff} \tag{1}$$

$$\sigma_{eff} = (\sigma_f - \sigma_{rev})/2 + \sigma^*/2 \tag{2}$$

where $\sigma_f$ is the flow stress related to unloading, $\sigma_{rev}$ is the reverse flow stress, and $\sigma^*$ is the stress from the point of unloading to the stage of recovery elastic deformation. Figure 10c is a schematic diagram of HDI stress calculation, and the above parameters are marked in Figure 10c. When the Cu volume fraction increased from 10% to 30%, the HDI stress of the Ni-Cu HS material continuously increased with the Cu content, and when the Cu volume fraction was further increased, the HDI stress of the Ni-Cu HS material gradually decreased with the increase in the Cu volume fraction. Figure 10d shows that as the strain increased, the HDI stress of the Ni-Cu heterostructure material also increased gradually.

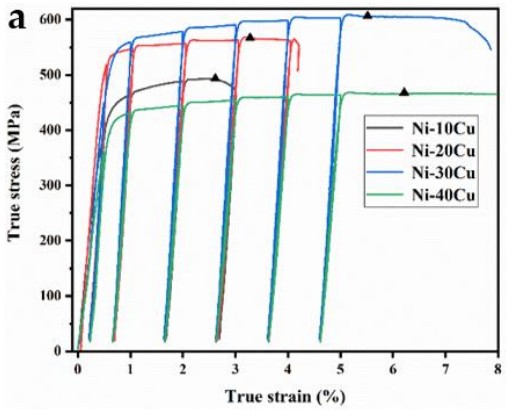

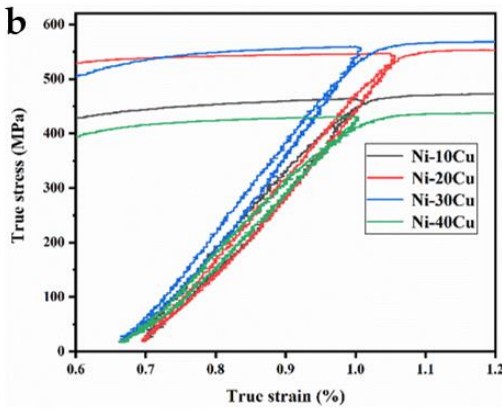

**Figure 10.** *Cont.*

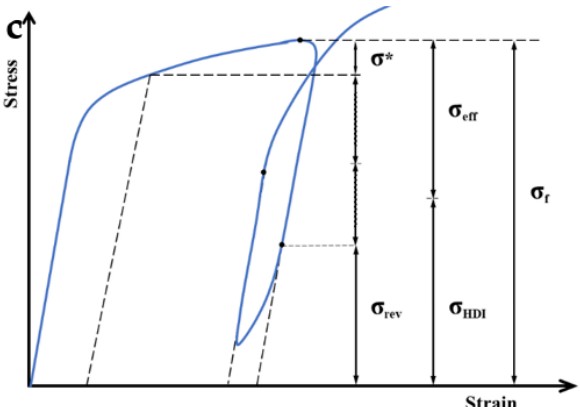
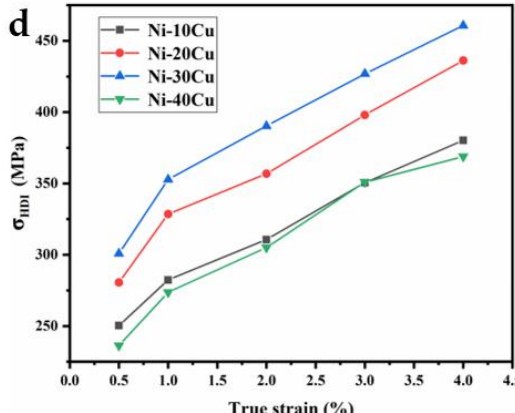

**Figure 10.** (**a**) LUR curve of Ni-10Cu, Ni-20Cu, Ni-30Cu, and Ni-40Cu; (**b**) partial enlarged view of the LUR curve; (**c**) schematic diagram showing the calculation of the HDI stress; (**d**) relationship between the back stress value and true strain, calculated according to Figure 10a.

## 4. Discussion

Rubio et al. [19] fabricated Ni-Cu heterostructures with gradient structures by SPS, confirming the possibility of Ni-Cu forming heterostructures. Song et al. [20] successfully synthesized a Ni-Cu bulk alloy with a maximum yield strength of 440 MPa by SPS. In this experiment, a Ni-30Cu heterostructure material was formed through rolling deformation and subsequent heat treatment, and the maximum yield strength could reach 556 MPa. The Ni-30Cu heterostructure material had a good combination of strength and plasticity. The higher yield strength may have been due to the combined effect of grain refinement due to rolling and the HDI strengthening effect.

During the rolling process, the material underwent severe plastic deformation, and the dislocation energy gradually increased. The Ni grains and Cu grains were elongated into slender fibrous grains along the RD, and the grains became smaller along the ND. After annealing at 450 °C for 1 h, the dislocation energy and grain boundary energy in the Cu region gradually decreased, and the long fibrous grains in the Cu region recrystallized into equiaxed grains. At this temperature, the Ni grains had not yet reached their recrystallization temperature, and therefore, the Ni grains were refined, but remained as elongated fibrous grains. When subjected to external force, the equiaxed grains in the Cu region could store more dislocations, and the Cu region could maintain the plasticity of the material, whereas the Ni region was annealed, the Ni grains were refined, and the Ni grains maintained the strength of the Ni-Cu HS materials.

When the Ni-Cu heterostructure material was subjected to external stress, the hard Ni domains underwent plastic deformation, whereas the soft Cu domains stayed in the elastic deformation stage. Due to the inconsistent deformation of the hard Ni domains and the soft Cu domains, a strain gradient was easily generated at the interface to maintain the interface balance, and the strain gradient was generated by the GNDs. The GNDs generated reverse stress in the Cu domain, that is, the strengthening effect of reverse stress; the Ni domain generated forward stress, which weakened the Ni region, and the forward stress and reverse stress together formed HDI. With the increase in the Cu volume fraction, the interface density of soft Cu domains and hard Ni domains increased gradually, and the strengthening effect of HDI at the Ni/Cu interface was stronger. Therefore, the yield strength of the material increased with the Cu volume fraction. However, when the volume fraction of Cu was too large, the reduction effect of yield strength brought by Cu exceeded that of the HDI strengthening, which indicates that the yield strength of Ni-Cu materials gradually decreased with the increase in the Cu volume fraction.

## 5. Conclusions

Ni-Cu HS materials are synthesized by powder metallurgy, rolling deformation, and subsequent heat treatment. In this paper, the relationship between the microstructure and mechanical properties of Ni-Cu heterostructures was studied. The main conclusions are as follows:

1.  In the Ni-Cu HS material, the Cu domain was uniformly distributed in the Ni region in the form of long fibers, and the Ni phase and the Cu phase were alternately and uniformly distributed to form a layered structure.
2.  In the sintered Ni-30Cu, the Ni grains and Cu grains were approximately equiaxed, and the Ni-30Cu with 80% deformation at room temperature was elongated into a sheet-like structure along the rolling direction. After annealing, the grains in the Cu region were recrystallized into equiaxed grains, and the grains in the Ni region were refined.
3.  The uniform elongation of Ni-Cu heterostructure materials increased gradually with the increase in the Cu volume fraction, and the yield strength first increased and then decreased gradually with the increase in the Cu volume fraction. The Ni-30Cu heterostructure material had a good combination of strength and plasticity.
4.  When the Cu volume fraction was less than 30%, the HDI strengthening effect in the Ni-Cu HS material could offset the effect of the yield strength reduction caused by Cu; with the further increase in the Cu volume fraction, the HDI strengthening effect was less than the yield strength reduction effect brought on by Cu.

**Author Contributions:** Conceptualization, Y.L.; methodology, W.A.; software, W.A.; validation, W.A. and Z.Y.; formal analysis, W.A. and Y.L.; investigation, W.A.; resources, Y.L.; data curation, Y.L.; writing—original draft preparation, W.A.; writing—review and editing, W.A. and Y.L.; visualization, Y.L.; supervision, Y.L.; project administration, Y.L.; funding acquisition, Y.L. All authors have read and agreed to the published version of the manuscript.

**Funding:** This research received no external funding.

**Institutional Review Board Statement:** Not applicable.

**Informed Consent Statement:** Not applicable.

**Data Availability Statement:** The raw and processed data required to reproduce these results are available by contacting the authors.

**Acknowledgments:** The present study was financially supported by the research funding from Wuhan University of Technology, China, for Newly Recruited Distinguished Professors (grant number: 471-40120281).

**Conflicts of Interest:** The authors declare no conflict of interest.

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
