# Peer review of "Synthesis of Ni-Cu Heterostructures with SPS to Achieve a Balance of Strength and Plasticity"

_metals, doi:10.3390/met12071093_

Round 1

Reviewer 1 Report

This paper is relatively novel and studies Ni-Cu HS material with different Cu contents. However, the content contrast in the article is not high, there is a lack of corresponding contrast experiments, and there are many obvious errors in the article. There are some errors in the article, such as Figure 1, which has a period after the title, while Fig. 2 does not, and the Figure1 is not abbreviated. There also need the SEM images and EDS spectra of Ni-Cu HS materials with different Cu contents both without treatment and after rolling and subsequent heat treatment.  The author is not serious enough. It is recommended to resubmit after modification.

Abstract

1.  Lines 13-14: This feature is missing from previous HS materials. Why Ni-Cu HS materials has this feature? It is first time to find the feature in the Ni-Cu HS materials? Rewrite this sentence. 

2. Line 21: Hetero-deformation induced hardening or Hetero-deformation induced  strengthening? Check.

Introduction

3. Line 45: Why appears the soft Mg domains? Should refer to soft Cu domains information.

4. The authors did not take into account in their article many important articles on thematic publications. This needs to be corrected. For example, Konovalov, S., Osintsev, K., Golubeva, A., Smelov, V., Ivanov, Y., Chen, X., Komissarova, I. Surface modification of Ti-based alloy by selective laser melting of Ni-based superalloy powder (2020) Journal of Materials Research and Technology, 9 (4), pp. 8796-8807. DOI: 10.1016/j.jmrt.2020.06.016; Guo, Y., Xu, Z., Liu, M., Zu, S., Yang, Y., Wang, Q., Yu, Z., Zhang, Z., Ren, L. The corrosion resistance, biocompatibility and biomineralization of the dicalcium phosphate dihydrate coating on the surface of the additively manufactured NiTi alloy (2022) Journal of Materials Research and Technology, 17, pp. 622-635. DOI: 10.1016/j.jmrt.2022.01.063; Xiang, K., Chai, L., Zhang, C., Guan, H., Wang, Y., Ma, Y., Sun, Q., Li, Y. Investigation of microstructure and wear resistance of laser-clad CoCrNiTi and CrFeNiTi medium-entropy alloy coatings on Ti sheet

(2022) Optics and Laser Technology, 145, 107518. DOI: 10.1016/j.optlastec.2021.107518

Material and methods

5. A schematic diagram of the sample is best given in the Materials and methods section. Because where is the metallographic structure taken from the sample, and where is the strength and toughness taken from the sample, don't know?

Results 

6. lines 109-110: After rolling and subsequent heat treatment, Cu was embedded in the Ni matrix in the form of long fibers. What is the growth relationship of untreated Cu and Ni?

7. Modify the Fig.4a, by the way mark which is Cu domains, and which is Ni domains.

8. Lines 249-250: where are the statistics of average grain size of Ni grains and Cu grains along the RD? Why the grain sizes are not same both in Fig.6a and Fig.6b, as they are both along the ND.

9. Lines 283-284: Why it is the Fig.6? It is Fig.7.

10. Lines 298-299: Check the sentence completeness.

11. Lines 316-317: It should be Fig.8.

12. Lines 317-318: According to a ?

Discussion

13.  Lines 327: How to determine the recrystallization temperature at this time? What to measure. 

Author Response

I have responded to your review comments, please see the attachment for details, thank you very much for your review.

Reviewer 2 Report

The manuscript of the article presents new experimental results concerning the influence of the parameters of Ni-Cu layered heterostructures on the mechanical properties of Ni-10Cu, Ni-20Cu and Ni-30Cu materials. The relevance of the study is due to the demand for a scientific basis for the creation of new materials with the required combination of deformation and strength properties achieved as a result of the mechanical response of heterostructures with a strength difference between soft and hard domains. Heterostructures in Ni-Cu  system were formed as a result of electrospark plasma sintering of powder mixtures, followed by rolling and heat treatment of the resulting volumes of sintered material. Data on the distribution of Ni and Cu in the heterostructures of Ni-10Cu, Ni-20Cu, and Ni-30Cu materials were obtained. The phase distribution maps for Ni-30Cu systems were determined at different stages of the formation of heterostructures during the preparation of the material. Data are obtained on the size distribution of Ni and Cu grains in rolled material samples. As a result of the experimental tests of the samples of the material obtained, it was found that the uniform elongation of the material with Ni-Cu heterostructures increases with an increase in the volume fraction of copper. The yield strength of the material under quasi-static loading, unloading and reloading under uniaxial tension conditions first increases and then gradually decreases with increasing volume fraction of copper. The Ni-30Cu material with a layered heterostructures and elongated copper fibers in the Ni region has a good combination of strength and ductility. The research results show that when the volume fraction of Cu is less than 30%, the strengthening effect in the Ni-Cu heterostructures of the material can compensate for the effect of lowering the yield strength caused by the presence of Cu in the material. It was found that with an increase in deformation during rolling of the Ni-30Cu sintered powder material; the grain structure with equiaxed sizes of Ni and Cu is transformed into a structure with elongated grains along the rolling direction. However, after annealing the rolled material, the grains in the Cu region are recrystallized into equiaxed grains, while the grains in the Ni region are refined. 

 The reliability of the results and conclusions is beyond doubt. The results presented in the manuscript of the article are of interest to specialists working in the field of creating materials with heterostructures, as well as to developers of computational models and methods for designing new materials and technologies for their production. 

The manuscript needs improvement.

 1) There are two different Figures in the manuscript with the same number 6 (Line 249 and Line 283). The numbering of the Figures should be changed. 

2) In Figure 6 (Line 283) it is necessary to clarify which stresses and which strains (True ones or Engineering ones) are shown in the Figure.

3) The strain rate and temperature of specimens testing should be indicated, the results of which are shown in Fig. 6 (Line 6) and Fig. 7.

Author Response

(The authors gave the same response as above.)

Reviewer 3 Report

The study analyzes the balance between strength and plasticity through spark plasma sintering (SPS) of Ni-Cu heterostructures (HSs). The processes of deformation by rolling and subsequent heat treatment have been used.
Although the study may present aspects of interest, I suggest important changes before publication.

Abstract
1. Please reword in the abstract what is the main contribution of this paper in comparison with the related literature. What is the novelty? What is new?

Introduction
2. I recommend that you develop in the introduction the contributions developed by other researchers.

Materials and methods
This section is not at all clear and there are some confusing parts. I recommend the development of a figure with a schematic diagram summarizing the stages of the experimental. This would make everything clearer.
I recommend the development of a table with each of the parameters used in the different tests performed.
It would be advisable to have an image or several images showing the development of the experimental and its measurements.

Results
In the line 81 it speaks of the figure 2a and 2b, in the figure 2 I do not appreciate that difference.
In line 158 it speaks of figure 4b, but not of figure 4a. Does it mean that this one is dispensable? If not, comment it.
There are two figures 6. Nothing is said about the first figure 6, why?
On line 254, figure 7 is mentioned, but then figure 6 is repeated. Is this a mistake?
The equations have not been numbered, they have not been referenced and their units are not indicated.
In line 288, figure 8 is mentioned and then figure 7 appears. Is this an mistake?

Discussion
No discussion has been developed with the work of other researchers. Please do so.

Conclusions
Check the spaces between the words. There is a lack of uniformity.

Refference
I would recommend the inclusion of the DOI.

Author Response

(The authors gave the same response as above.)

Round 2

Reviewer 1 Report

Authors have made corrections. I dont comments.

Reviewer 3 Report

The changes suggested to the authors have been made. The document is accepted with the current format.